# Testing structural identifiability by a simple scaling method

**Mario Castro** [1,2]⊛*, **Rob J. de Boer** [3]⊛

**1** Grupo Interdisciplinar de Sistemas Complejos (GISC), Madrid, Spain, **2** Instituto de Investigación Tecnológica (IIT), Universidad Pontificia Comillas, Madrid, E28015, Spain, **3** Theoretical Biology and Bioinformatics, Utrecht University, Utrecht, The Netherlands

⊛ Both authors also contributed equally to this work.
* marioc@comillas.edu

**Data Availability Statement:** All relevant data are within the manuscript and its Supporting information files.

**Funding:** This work was funded by Agencia Estatal de Investigación (FIS2016-78883-C2-2-P, PID2019-106339GB-I00) to MC. The funders had

## Abstract

Successful mathematical modeling of biological processes relies on the expertise of the modeler to capture the essential mechanisms in the process at hand and on the ability to extract useful information from empirical data. A model is said to be structurally unidentifiable, if different quantitative sets of parameters provide the same observable outcome. This is typical (but not exclusive) of partially observed problems in which only a few variables can be experimentally measured. Most of the available methods to test the structural identifiability of a model are either too complex mathematically for the general practitioner to be applied, or require involved calculations or numerical computation for complex non-linear models. In this work, we present a new analytical method to test structural identifiability of models based on ordinary differential equations, based on the invariance of the equations under the scaling transformation of its parameters. The method is based on rigorous mathematical results but it is easy and quick to apply, even to test the identifiability of sophisticated highly non-linear models. We illustrate our method by example and compare its performance with other existing methods in the literature.

## Author summary

Theoretical Biology is a useful approach to explain, generate hypotheses, or discriminate among competing theories. A well-formulated model has to be complex enough to capture the relevant mechanisms of the problem, and simple enough to be fitted to data. Structural identifiability tests aim to recognize, in advance, if the structure of the model allows parameter fitting even with unlimited high-quality data. Available methods require advanced mathematical skills, or are too costly for high-dimensional non-linear models. We propose an analytical method based on scale invariance of the equations. It provides definite answers to the structural identifiability problem while being simple enough to be performed in a few lines of calculations without any computational aid. It favorably compares with other existing methods.

This is a *PLOS Computational Biology* Methods paper.

no role in study design, data collection and analysis, decision to publish, or preparation of the manuscript.

**Competing interests:** The authors have declared that no competing interests exist.

## Introduction

Mathematical models contribute to our understanding of Biology in several ways ranging from the quantification of biological processes to reconciling conflicting experiments [1]. In many cases, this requires formulating a mathematical model and extracting quantitative estimates of its parameters from the experimental data. Parameters are typically unknown constants that change the behavior of the model. While it is usually recognized that parameter estimation requires the availability of sufficient informative data, sometimes it is not possible to estimate all parameters due to the structure of the model (whatever the quantity or quality of the data), even with large amounts of noiseless observations. This inability is referred to as 'structural identifiability', a concept introduced decades ago by Bellman and Åström [2, 3], as opposed to the 'practical identifiability' that depends on limitations set by the data. Practical identifiability has important consequences that can lead to questionable interpretations of the data leading to some recent controversy around this point [4, 5]. Structural identifiability poses an unsolvable limitation as it is unrelated to the resolution of the experimental data collection or the number of observations.

Structural identifiability is a necessary condition for model fitting and should be used before any attempt to extract information about the parameters, and as a test of the applicability of the model itself. Importantly, the quality of the fit does not guarantee that the estimated parameters are meaningful. In practice, this is both uncontrolled and misleading, as many fitting tools provide information about the goodness of fit but do not check sensitivity or identifiability. Structural identifiability can be qualified as global or local [6–10]. Global structural identifiability tests the ability to estimate unique sets of parameters, while local (or simply, structural identifiability) means that parameters can be estimated only in a limited subset of the space of parameters. In practical terms, these definitions can be translated into the language of sensitivity analysis as identifiability requires that (i) the columns of the sensitivity matrix are linearly independent, and (ii) each of its columns has at least one large entry [11, 12].

Traditionally, work primarily focused on linear systems [2, 3, 13] based on ordinary differential equations (ODE). For non-linear models, those methods cannot be applied, so many methods have been proposed in the literature to address structural identifiability. Early attempts were based on power series expansions of the original non-linear system [14], the similarity transformation method [15–17] or the so-called direct-test method proposed by Denis-Vidal and Joly-Blanchard [18, 19]. These methods exploit the definition of identifiability either analytically [18] or numerically [20–25], but they are not generically suitable for high-dimensional problems. Xia and Moog [6, 26] proposed an alternative to these classical methods based on the implicit function theorem, but this method also becomes involved to apply for complex models [27].

Another approach that is becoming mainstream is based on the framework of differential algebra [28–31]. These methods are also difficult to apply, requiring advanced mathematical skills and, in some cases, replace highly non-linear terms by polynomial approximations that simplify the analysis. On the positive side, they are based on rigorous mathematical theories, are suitable for non-linear models and, more importantly, they can be coded using existing symbolic computational libraries. In this regard, it is worth mentioning DAISY [32], GenSSI [33], COMBOS [34] or, more recently, SIAN [35].

In almost all cases, the major disadvantage of these methods is their difficulty to apply them to even a few differential equations, hence requiring advanced mathematical skills and/or dedicated numerical or symbolic software (that is frequently unable to handle the complexity of the problem). This explains why, despite the huge volume of publications in the field of

theoretical biology, only a few address parameter identifiability explicitly. In this paper, we introduce a simple method to assess local structural identifiability of ODE models that reduces the complexity of existing methods and can bring identifiability testing to a broader audience. Our method is based on simple scaling transformations, and the solution of simple sparse systems of equations. Identifiability for stochastic models [36] is out of the scope of our work.

## Method

### A couple of motivating examples

Consider a simple *death* model in which the death rate is the product of two parameters $\lambda_1$ and $\lambda_2$, namely

$$\frac{dx}{dt} = -\lambda_1 \lambda_2 x, \qquad x(0) = x_0, \tag{1}$$

with the solution

$$x = x(\lambda_1, \lambda_2, t) = x_0 e^{-\lambda_1 \lambda_2 t}. \tag{2}$$

It is evident that from an experiment only the product $\lambda_1 \lambda_2$ can be inferred, and not any of the two independently. Following the 'actionable' definition in Ref. [11], local structural identifiability is directly linked to the linear independence of the columns of the sensitivity matrix, $S_{ij}$, of the variable $x_i$ with respect to parameter $\lambda_j$

$$S_{ij}(x_1, \ldots, x_r, x_{r+1} \ldots x_n; \lambda_1, \ldots, \lambda_m) \equiv \frac{\partial x_i}{\partial \lambda_j} \tag{3}$$

Here, we will work with a related (dimensionless) quantity called the relative sensitivity, or simply the elasticity matrix $K$ with elements $K_{ij}$ given by

$$K_{ij}(x_1, \ldots, x_r, x_{r+1} \ldots x_n; \lambda_1, \ldots, \lambda_m) \equiv \frac{\partial \log x_i}{\partial \log \lambda_j} = \frac{\lambda_j}{x_i} \frac{\partial x_i}{\partial \lambda_j} = \frac{\lambda_j}{x_i} S_{ij}. \tag{4}$$

The *logarithm* in the definition of the elasticity matrix provides a clear-cut interpretation of its coefficients. Thus, if $K_{ij} = 1$, a 10% increase in $\lambda_j$ implies a 10% increase in $x_i$, and if $K_{ij} = 0.5$, that very same increase in $\lambda_j$ translates only to a 5% increase in $x_i$.

For Eq (1), the elasticity matrix would be simply a $1 \times 2$ matrix,

$$K = (K_{11} \ K_{12}),$$

with

$$K_{11} \equiv \frac{\lambda_1}{x} \frac{\partial x}{\partial \lambda_1} \ , \text{and} \quad K_{12} \equiv \frac{\lambda_2}{x} \frac{\partial x}{\partial \lambda_2} \ . \tag{5}$$

We now propose to multiply $\lambda_1$ with a generic scale factor $u$, and to divide $\lambda_2$ by the same factor, such that the solution remains invariant. Deriving the scaled solution of Eq (2) with respect to that scale factor $u$, and by the chain rule,

$$\frac{dx}{du} = 0 \ (\text{as } u \text{ is arbitrary}) \tag{6}$$

and, also,

$$\frac{dx(u\lambda_1, \lambda_2/u, t)}{du} = \frac{\partial x}{\partial \lambda_1}\lambda_1 - \frac{\lambda_2}{u^2}\frac{\partial x}{\partial \lambda_2} = 0 \tag{7}$$

where the last equality follows from Eq (6)

Rearranging Eq (7) and dividing by $x$,

$$\frac{\lambda_1}{x}\frac{\partial x}{\partial \lambda_1} = \frac{\lambda_2}{u^2 x}\frac{\partial x}{\partial \lambda_2} \Rightarrow K_{11} = \frac{1}{u^2}K_{12}, \tag{8}$$

so both *columns* of the elasticity matrix are linearly dependent and, accordingly, $\lambda_1$ and $\lambda_2$ are unidentifiable. In this particular case, the exact solution confirms this result:

$$K_{11} = K_{12} = \lambda_1\lambda_2 t.$$

In this case we had complete knowledge of the solution, and consequently, it was straightforward to find the right way to introduce the scaling $u$. Fortunately, this simple scaling calculation can also be performed directly on Eq (1). Introducing two unknown scaling factors, $u_1$ and $u_2$, into that equation,

$$\frac{dx}{dt} = -u_1\lambda_1 u_2\lambda_2 x \ .$$

Requiring that this remains identical (or, more formally, *invariant*) to Eq (1), i.e., $\lambda_1\lambda_2 \, x = u_1\lambda_1 u_2\lambda_2 x$, we conclude that $u_1 u_2 = 1$. The fact that $u_1$ and $u_2$ cannot be solved individually, also means that the real values of $\lambda_1$ and $\lambda_2$ cannot be determined, namely both parameters are unidentifiable.

Next consider a death model with immigration:

$$\frac{dx}{dt} = \lambda_1 - \lambda_2 x. \tag{9}$$

In this case, to leave the system invariant we need to find $u_1$ and $u_2$ such that

$$\lambda_1 - \lambda_2 x(t) = u_1\lambda_1 - u_2\lambda_2 x(t)$$

for all values of $x$ at any time. Rearranging the latter equation,

$$(1 - u_1)\lambda_1 = (1 - u_2)\lambda_2 x(t),$$

where the left-hand side of the last equation is a constant and the right-hand side depends on time. Hence the only possible solution to the latter equation is $u_1 = u_2 = 1$ implying that both $\lambda_1$ and $\lambda_2$ are locally identifiable. Notice the difference with the preceding case, Eq (1), in which an infinite number of combinations of the scaling factors satisfy the invariance condition.

These simple examples illustrate how scaling invariance of the model equations can be used to determine whether the parameters are unidentifiable or not. We prove this result more rigorously in S1 Text.

## Description of the method

Let us define a general ODE model characterized by the time evolution of $n$ variables, $x_i(t)$, depending on $m$ parameters $\lambda_j$,

$$\frac{dx_i}{dt} = f_i(x_1, \ldots, x_r, x_{r+1} \ldots x_n; \lambda_1, \ldots, \lambda_m) \qquad i = 1, \ldots, n \; , \tag{10}$$

$$x_i(0) = x_{i,0}, \qquad i = 1, \ldots, n \; , \tag{11}$$

where the functions $f_i$ depend on the specific details of the problem at hand and $x_{i,0}$ are the initial conditions. We need to distinguish between those variables that can be observed (measured) in the experiment, $x_1 \ldots x_r$, and those which cannot (they are often referred to as *latent* variables), $x_{r+1} \ldots x_n$.

As we will prove below, the simplicity of our method relies on the ability to decompose the functions $f_i$ as a sum of $M$ functional independent *summands*, $f_{ik}$,

$$f_i(x_1, \ldots, x_r, x_{r+1} \ldots x_n; \lambda_1, \ldots, \lambda_m) = \sum_{k=1}^{M} f_{ik}(\tilde{x}_k, \tilde{\lambda}_k), \tag{12}$$

having the property that $f_{ik}$ is functionally independent of $f_{il}$ for every $k \neq l$. For brevity, $\tilde{x}_k, \tilde{\lambda}_k$ denote the subset of variables and parameters included in the function $f_{ik}$.

The notion of linear independent functions and how to test it is summarized in S1 Text. However, a simple definition would be: If $f_1(x_1, x_2, \ldots), \ldots, f_n(x_1, x_2, \ldots)$ are linearly independent functions, then the only solution of the equation

$$\sum_{i=1}^{n} a_i f_i(x_1, x_2, \ldots) = 0 \tag{13}$$

is $a_1 = \ldots = a_n = 0$.

Typical examples of functionally independent functions are summarized in Table 1. For instance, $f_{11} = ax_1, f_{12} = bx_1x_3, f_{13} = (c + x_4)^{-1}$ are functionally independent, whereas examples of dependent functions would be $f_{11} = ax_1x_2$ and $f_{12} = bx_1x_2$. Note that it is not required that $f_{ij}$ and $f_{kj}$ are independent (as they appear in different equations). For instance, in the example in Eq (9) can be decomposed in polynomials of degree 0 (a constant) and 1 (a linear function), namely

$$f_{11} = \lambda_1 \; , \text{and} \quad f_{12} = -\lambda_2 x \; .$$

We summarize our method in Box 1.

**Table 1. A collection of frequent linear independent functions: All the functions listed in the Table are independent to each other (of the same or different type).** We assume that $\lambda_1 \neq \lambda_2$ in all of the cases.

| Type | Examples |
|---|---|
| Polynomial (one variable) | $x^0, x, x^2, x^3, \ldots$ |
| Polynomial (more than one variable) | $x_1x_2, x_1^2x_2, x_1x_2x_3, \ldots$ |
| Rational | $\frac{1}{\lambda+x_1}, \frac{x_1}{\lambda+x_1}, \frac{1}{x_1+x_2}, \frac{1}{\lambda_1+x_1+x_2}, \frac{1}{(\lambda+x_1)^2}, \ldots$ |
| Exponential | $e^{\lambda_1 x_1}, e^{\lambda_2 x_1}, e^{\lambda_1 x_2}$ |
| Sigmoid | $\frac{1}{\lambda_1+e^{-\lambda_1 x_1}}, \frac{1}{\lambda_1+e^{-\lambda_1 x_2}}, \frac{1}{\lambda_1+e^{-\lambda_2 x_1}}, \frac{1}{(\lambda_1+e^{-\lambda_1 x_1})^2}, \ldots$ |
| Trigonometric | $\sin \lambda_1 x_1, \sin \lambda_1 x_2 \sin \lambda_2 x_1, \cos \lambda_1 x_1, \tan \lambda_1 x_2, \ldots$ |

Box 1: Summary of the scale invariance local structural identifiability method introduced in this work

1. Scale all parameters and all unobserved variables by unknown scaling factors, $u$:

$$\lambda_i \rightarrow u_{\lambda_i} \lambda_i \quad i = 1, \dots, m$$

$$x_j \rightarrow u_{x_j} x_j \quad j = r+1, \dots, n$$

and substitute them into Eq (15) below.

2. Equate each functionally independent function, $f_{ik}$, to its scaled version. Namely,

$$f_{ik}(\tilde{x}, \tilde{\lambda}) = \frac{1}{u_{x_i}} f_{ik}(u_{\tilde{x}} \tilde{x}, u_{\tilde{\lambda}} \tilde{\lambda}) \tag{14}$$

where $u_{x_i} = 1$ for $1 \leq i \leq r$ and the prefactor in the right-hand side of the equation comes from the scaling of $\frac{dx_i}{dt} \rightarrow u_{x_i} \frac{dx_i}{dt}$. From Eq (11) it follows that $u_{x_i} = u_{x_{i,0}}$.

3. From Eq (14), find combinations of the scaling factors $u$ that leave the system invariant. Hereafter, we will denote these as the **identifiability equations** of the model (see Eq (24) below).

4. Only the parameters $\lambda_i$ with a solution $u_{\lambda_i} = 1$ are identifiable. Only the variables, $x_i$ with $u_{x_i} = 1$ are observable. Otherwise, parameters whose scaling factors are coupled, form identifiable groups but cannot be identified independently.

In summary, our method reduces the complexity of finding identifiable parameters to finding which scaling factors do not satisfy the trivial solution $u_i = 1$. In the literature, when a scaling factor is related to one of the latent variables $x_{r+1} \dots x_n$, if $u_{x_k} = 1$, then $x_k$ is said to be *observable* [10]. Thus, our method addresses at the same time identifiability and observability. Additionally, irreducible equations involving two or more parameters provide the so-called identifiable groups of variables that cannot be fitted independently. In the case of the pure death model above, the identifiability equation $u_{\lambda_1} u_{\lambda_2} = 1$ is a signature of the unidentifiable group $\lambda_1 \lambda_2$. This is interesting as groups involving latent variables (for instance, $u_{x_j} u_{\lambda_k}$) would inform future experiments aimed to observe that variable and decouple that group.

It is also worth mentioning that our identifiability test (illustrated by example in S1 Text) provides a simple way to find a type of symmetry that is related to scale invariance. More sophisticated methods have been introduced in the literature to address other symmetries [37–39] using the theory of Lie group transformations, however, that approach involves complex calculations assisted by symbolic computations.

## Results

### The main result

Now we are equipped to prove the main result of the paper. We will proceed in two steps: firstly, we will show how Eq (14) is translated into a set of equations for the scaling factors $u$.

Secondly, we will connect the elasticity matrix with the solution of the identifiability equations and the identifiability of the parameters.

Consider a model described by a set of $n$ ordinary differential equations (ODE)

$$\frac{dx_i}{dt} = f_i(x_1, \ldots, x_r, x_{r+1} \ldots x_n; \lambda_1, \ldots, \lambda_m) = \sum_{k=1}^{M} f_{ik}(\tilde{x}_k, \tilde{\lambda}_k), \tag{15}$$

where $f_{ik}$ is functionally independent of $f_{il}$ for every $k \neq l$ (namely, they satisfy the generalized Wronskian theorem; see S1 Text). For the sake of simplicity, we denote $\tilde{x}_k$ and $\tilde{\lambda}_k$ the subset of variables and parameters of function $f_{ik}$.

Motivated by Eqs (1)–(5), we seek for scaling of the parameters that leave the system invariant. As we prove below, this invariance (or lack of) is related to the identifiability of the parameters. Hence, if we define the following scaling transformation:

$$\lambda_i \rightarrow u_{\lambda_i} \lambda_i, \quad i = 1, \ldots, m \qquad x_j \rightarrow u_{x_j} x_j, \quad j = r+1, \ldots, n \tag{16}$$

(where the variables $x_1 \ldots x_r$ are unmodified as we can measure them in the experiment) we can write the following set of re-scaled equations:

$$\frac{dx_i}{dt} = \sum_{k=1}^{M} f_{ik}(u_{\tilde{x}_k} \tilde{x}_k, u_{\tilde{\lambda}_k} \tilde{\lambda}_k), \ i = 1, \ldots, r \tag{17}$$

$$x_i = x_{i,0}, \ i = 1, \ldots, r \tag{18}$$

$$u_{x_i} \frac{dx_i}{dt} = \sum_{k=1}^{M} f_{ik}(u_{\tilde{x}_k} \tilde{x}_k, u_{\tilde{\lambda}_k} \tilde{\lambda}_k), \ i = r+1, \ldots, n \tag{19}$$

$$u_{x_i} x_i = u_{x_{i,0}} x_{i,0}, \ i = r+1, \ldots, n \tag{20}$$

where $M$ is the number of functional independent summands in the equation. It is convenient to rewrite Eq (19) as

$$\frac{dx_i}{dt} = \frac{1}{u_{x_i}} \sum_{k=1}^{M} f_{ik}(u_{\tilde{x}_k} \tilde{x}_k, u_{\tilde{\lambda}_k} \tilde{\lambda}_k), \ i = r+1, \ldots n \tag{21}$$

to perform the scale invariance analysis below in a simpler way.

If the solution is invariant under this transformation, then the right-hand sides of Eq (15) and, consequently Eqs () should be equal. Besides, by the functional linear independence of the functions $f_{ik}$ we can split each summand. Thus,

$$f_{ik}(\tilde{x}_k, \tilde{\lambda}_k) = f_{ik}(u_{\tilde{x}_k} \tilde{x}_k, u_{\tilde{\lambda}_k} \tilde{\lambda}_k), \quad i = 1, \ldots, r \tag{22}$$

and

$$f_{ik}(\tilde{x}_k, \tilde{\lambda}_k) = \frac{1}{u_{x_i}} f_{ik}(u_{\tilde{x}_k} \tilde{x}_k, u_{\tilde{\lambda}_k} \tilde{\lambda}_k), \quad i = r+1, \ldots, n \tag{23}$$

This new set of equations is much easier to solve than the one that we would obtain from Eqs (17)–(19) (which would be equivalent to the so-called direct-test method [18]). Eqs (22) and (23) admit the trivial solution $u_{\tilde{x}_k} = u_{\tilde{\lambda}_k} = 1$. Alternatively, some of the parameters are

functionally related to each other. Generically, they can be written as

$$u_{\lambda_k} = F_k(u_{m_1}, u_{m_2}, \ldots), \tag{24}$$

Note that, for each parameter $k$, the scaling $u_{\lambda_k}$ will depend only on a subset of all the scaling factors $m_1, m_2, \ldots$ We denote Eq (24) the **identifiability equations** of the model. A third possibility would be that some scaling factors take fixed values different from 1. We discuss that case below.

Let us now connect the identifiability equations with the concept of local structural identifiability. If we take the partial derivative of the following (invariant equation)

$$x_i(x_1, \ldots, x_r, u_{x_{r+1}} x_{r+1} \ldots u_{x_n} x_n; u_{\lambda_1} \lambda_1, \ldots, u_{\lambda_m} \lambda_m) = x_i(x_1, \ldots, x_r, x_{r+1} \ldots x_n; \lambda_1, \ldots, \lambda_m)$$

with respect to $u_{\lambda_k}$, by the chain rule, it follows that

$$\frac{\partial x_i}{\partial \lambda_k} \lambda_k + \frac{\partial x_i}{\partial m_1} m_1 \beta_{m_1 k} + \frac{\partial x_i}{\partial m_2} m_2 \beta_{m_2 k} + \ldots = 0 \tag{25}$$

where, for convenience, we have defined

$$\beta_{mk} \equiv \frac{\partial u_m}{\partial u_{\lambda_k}} = \left( \frac{\partial F_k}{\partial u_m} \right)^{-1}.$$

Finally, dividing Eq (25) by $x_i$:

$$K_{ik} + \beta_{m_1 k} K_{im_1} + \beta_{m_2 k} K_{im_2} + \ldots = 0, \tag{26}$$

where $K_{im}$ are the elements of the elasticity matrix defined in Eq (4). Eq (26) implies that $K_{ik}$ can be written as a linear combination of other column(s) of the elasticity matrix. According to our discussion in the Introduction (see also Refs. [11, 12]) this is means that $\lambda_k$ is not identifiable.

Summarising, for each parameter $\lambda_k$ either $u_{\lambda_k} = 1$ or it is not identifiable. The adjective "local" follows because the method stems on the continuity of the derivative of $x_i(t)$ with respect to $\lambda_k$ to derive Eq (25). Thus, it is unable to capture any discrete transformations like, for instance,

$$\left\{ u_c \to 1, u_\delta \to 1, \qquad \left\{ u_c \to \frac{\delta}{c}, u_\delta \to \frac{c}{\delta} \right. \right.$$

discussed for Model 8 in S1 Text and that, as we anticipated above, is the third possible solution of the identifiability Eq (24).

### Example: An unidentifiable nonlinear model [16]

Here we show how to apply our method to a nonlinear model introduced in Ref. [16] (this model is mathematically equivalent to Model 2 in S1 Text).

$$\dot{x}_1 = \lambda_1 x_1^2 + \lambda_2 x_1 x_2, \tag{27}$$

$$\dot{x}_2 = \lambda_3 x_1^2 + \lambda_4 x_1 x_2, \tag{28}$$

$$x_1(0) = 0, \tag{29}$$

$$x_2(0) = 0, \tag{30}$$

$$x_1 \text{ is observed} \tag{31}$$

Following Box 1:

1. We re-scale the non-observed variables and parameters:

$$\begin{cases} x_2 & \to & u_{x_2} x_2 \\ \lambda_1 & \to & u_{\lambda_1} \lambda_1 \\ \lambda_2 & \to & u_{\lambda_2} \lambda_2 \, , \\ \lambda_3 & \to & u_{\lambda_3} \lambda_3 \\ \lambda_4 & \to & u_{\lambda_4} \lambda_4 \end{cases} \tag{32}$$

   as $x_1$ is observed (so, $u_{x_1} = 1$).

2. We define the functional linear independent functions:

$$f_{11} = \lambda_1 x_1^2 \qquad f_{12} = \lambda_2 x_1 x_2 \qquad f_{21} = \lambda_3 x_1^2 \qquad f_{22} = \lambda_4 x_1 x_2,$$

   and from Eq (14)

$$u_{\lambda_1} \lambda_1 x_1^2 = \lambda_1 x_1^2 \qquad u_{\lambda_2} u_{x_2} \lambda_2 x_1 x_2 = \lambda_2 x_1 x_2$$

   and

$$\frac{u_{\lambda_3}}{u_{x_2}} \lambda_3 x_1^2 = \lambda_3 x_1^2 \qquad u_{\lambda_4} \lambda_4 x_1 x_2 = \lambda_4 x_1 x_2$$

   respectively.

3. Manipulating the previous equations:

$$u_{\lambda_1} \lambda_1 \cancel{x_1^2} = \lambda_1 \cancel{x_1^2} \qquad u_{\lambda_2} u_{x_2} \lambda_2 \cancel{x_1 x_2} = \lambda_2 \cancel{x_1 x_2}$$

   and

$$\frac{u_{\lambda_3}}{u_{x_2}} \lambda_3 \cancel{x_1^2} = \lambda_3 \cancel{x_1^2} \qquad u_{\lambda_4} \lambda_4 \cancel{x_1 x_2} = \lambda_4 \cancel{x_1 x_2}$$

Hence, the identifiability equations are

$$\begin{cases} u_{\lambda_1} = 1 \\ u_{\lambda_2} u_{x_2} = 1 \\ u_{\lambda_3} = u_{x_2} \\ u_{\lambda_4} = 1 \end{cases} \tag{33}$$

4. As the system has more than 1 solution besides the trivial ($u_{\lambda_1} = u_{\lambda_2} = \ldots = 1$) it follows that the model is unidentifiable. Moreover, Eq (33) allows one to conclude that (i) if $x_2$ were to be observed ($u_{x_2} = 1$), all the parameters would be identifiable, and (ii) the combination $u_{\lambda_2} u_{\lambda_3}$ is identifiable as, for any *scale* of $x_2$, the condition $u_{\lambda_2} u_{\lambda_3} = 1$ is always fulfilled and hence $\lambda_2 \lambda_3$ is an identifiable group.

## Comparison with other methods

We have applied the method outlined in Box 1 to 13 different models defined and analyzed in detail in S1 Text. The choice is based on two criteria: on the one hand, models 1-5 are included for *pedagogical* purposes. They are simple enough to illustrate the novel method and most of the existing methods also provide the same definite answers. Models 6-13 were chosen because they have previously been analyzed using the methods summarized in the Introduction and in Table 2. This allows us to put our method in direct competition with those methods and to highlight their merits and limitations.

The results of this comparison are summarized in Table 3, which is an extension of a similar table in Ref. [7]. The column *Not Conclusive/Not Applicable* groups different situations in which a particular method do not provide a conclusive answer (or no answer at all). In general, it captures the fact that many of these methods are computationally demanding (after several hours they do not provide any answer) or that the computations do not converge numerically. For instance, in some implementations of the Differential Algebra method [32], when the number of observables is lower than the number of parameters, the computation requires the evaluation of high-order derivatives of the functions $f_i$ in Eq (11) what can be computationally prohibitive. In other cases, some criterion of applicability is not fulfilled (for instance, the observability rank condition for the similarity transformation method) or the method cannot

**Table 2. List of current methods testing structural identifiability.** We introduce here the acronyms referred to in Table 3.

| Method | Acronym | Main Ref. | Pros | Cons |
|---|---|---|---|---|
| Direct test method | DT | [18, 20] | Simple | Limited |
| Implicit function theorem | IFT | [26] | Software | Limited |
| Taylor series approach | TS | [14] | Simple | Computationally Expensive |
| Generating series approach | GS | [13] | Simple, Software | Computationally Expensive |
| Similarity Transformation | ST | [16] | Software | Computationally Expensive |
| Differential algebra | DA | [29, 32, 34] | Software, Conclusive | Limited, Comp. Expensive |
| Reaction Network theory | RNT | [40, 41] | Simple, Hybrid with other | Only reaction systems |
| STRIKE-GOLDD | SG | [9, 22] | Powerful, Software | Computationally Expensive |
| Scaling Invariance Method | **SIM** | This work | Simple, Widely applicable | Only Local Identifiability |

be solved if it involves the solution of a high-degree polynomial or transcendental equations (Direct Test method). These limitations are summarized succinctly in the *Cons* column in Table 2.

## Discussion and conclusions

Table 3 shows that our method can handle any complex model and provides a local structural identifiability criterion that is compatible with those methods capable of producing an answer. Thus, our method is widely applicable. It is worth noting that in several cases where our scaling method comes with a conclusive answer, other more complicated methods cannot address those cases (rightmost column in the table). As any global structural identifiable model is also local, our results are compatible with those methods that can address that difference.

Table 3 also highlights the huge discrepancies among methods. These conflicting conclusions are rather discomforting and deserve deeper clarification. The main source of conflict arises when comparing the Taylor series and the Generating series methods, as they transform the original problem into an approximate one. Also, they incorporate (rightly) the initial conditions into the computation while some implementations of the Differential algebra (DA) method do not (see the DAISY implementation [32]), what can lead to different conclusions. Regarding the DA method, in some instances random values are used for the parameters to handle the complexity of some models what, if those parameters are not properly explored, can lead to wrong conclusions.

So overall, we can distinguish three sources of discrepancy: local vs global structural identifiability (which is not an incompatibility as Global implies Local and our method is restricted to the latter); conclusive vs not conclusive (which favors our method as it is not limited by any computational constraint) and; the most concerning, incompatible conclusions. Here, our method is compatible with the conclusions of DA and hybrid methods such as Reaction network theory or STRIKE-GOLDD. As we mentioned in the introduction, Differential Algebra methods (and extensions) are considered the most reliable (when computable) and our method either agrees, or provides an answer where the other methods cannot. The discrepancies with other methods are due to limitations or uncontrolled approximations when applied to complex problems and have been already raised by other authors [7].

**Table 3. Summary of models compared in the literature: The number in brackets in the Model Name column corresponds to the number of observed variables.** Model Numbers correspond to those in Table A in S1 Text. The acronyms for the methods are summarized in Table 2. This table is an extension of Table 1 in Ref. [7].

| Model name | Main Ref. | Model Number | Global Struct. Id. | Local Struct. Id. | Unidentifiable | Not Conclusive Not Applicable |
|---|---|---|---|---|---|---|
| Goodwin model (1) | [7] | 6 | | | SG,**SIM** | TS,GS,ST,DT,DA,IFT,RNT |
| Goodwin model (all) | [7] | 6bis | | TS,GS,IFT,RNT | DA,SG,**SIM** | ST,DT |
| Circadian clock model | [42] | 7 | | | TS,GS,RNT,SG,**SIM** | ST,DT,DA,IFT |
| HIV model (1) | [6, 43] | 8 | | | All | |
| HIV model (2) | [6, 43] | 8bis | DA,IFT,RNT | TS,GS,**SIM** | | DT,ST |
| Linear HIV model (1) | [6, 43, 44] | 8ter | DA,IFT,RNT,SG | DT,ST,TS,GS,**SIM** | | |
| Glycolysis model | [45] | 9 | GS,DA,RNT | TS,**SIM** | | ST,DT |
| High dimensional model | [42] | 10 | TS,GS,DA,RNT | IFT,**SIM** | | ST,DT |
| NF-$\kappa$ model B (1) | [46] | 11 | | | SG, **SIM** | TS,GS,ST,DT,DA,IFT,RNT |
| NF-$\kappa$ model B (2) | [46] | 11bis | GS,RNT | TS,**SIM** | SG | ST,DT,DA,IFT |
| Pharmacokinetics model (1) | [47] | 12 | | TS,GS,RNT,SG,**SIM** | | ST,DT,DA,IFT |
| Pharmacokinetics model (2) | [47] | 12bis | DA | GS,SG,**SIM** | | ST,DT,IFT,RNT |
| Within-host virus model | [27] | 13 | DA | **SIM** | | TS,GS,ST,DT,IFT,RANT |

From viewpoint of performance, it is worth emphasizing that we have performed our test by hand, as illustrated in S1 Text, and that, after some practice (and using some interesting *motifs* as having sums of different parameters, or the coefficients related to *diagonal* terms in the system of equations) the calculations can be made in a few minutes. This contrasts with the most sophisticated methods that, by hand, can fill several pages [27] or take hours using symbolic computation packages.

Together, broad applicability and simplicity are the main signatures of our method and this may attract the interest of mathematical modelers and spread the *culture* of checking structural identifiability as a mandatory step when fitting experimental data.

We would like to highlight a connection with the so-called Buckingham-$\Pi$ theorem of dimensional analysis [48]. In some sense, the scale invariance property is related to the principle of dimensional homogeneity, *i.e.*, the constraints on the functional form of the independent variables with the parameters. Our identifiability equations are therefore similar to finding the so-called $\Pi$-groups in the theorem.

A limitation of the method is that it is restricted to testing local identifiability. This is implicit in the differentiability of the elasticity matrix which, by definition, is a local operation. Discrete symmetries are not captured, and more sophisticated methods (based on Lie group transformations [39]) are required. However, simple manipulation of the equations to remove the latent variables can improve the explanatory power of the method and might capture those discrete symmetries (see Sec. 3.8 of S1 Text). We leave that extension for future developments.

Finally, in this work we have chosen to solve the scaling factor equations directly as it is easy to perform with pen and paper. However, if we were to redefine the scaling factors as $u_i = e^{w_i}$, the new factors $w_i$ would obey a linear system of homogeneous equations. It is therefore expected that the problem of identifiability is related to the rank of the matrix defining the linear system of equations. In that regard, the theorems presented in S1 Text could be supplemented with generic results on homogeneous systems of equations. Thus, our results provide a solid ground for the method and indicate a venue for further development in other systems like delay-differential or partial differential equations.

Another open question is the identifiability problem of mixed-effect models, where parameters are not fixed quantities for each observation but, rather, they are drawn from a *meta-distribution* linking different subjects [49]. For instance, if one considers the simple model

$$\dot{x} = (a + b)x,$$

$a$ and $b$ are not identifiable. However, if they are assumed to be drawn from, say, two exponential distributions with *different* means $\mu_a$ and $\mu_b$, then the joint distribution for $\lambda \equiv a - b$ is given by

$$p(\lambda; \mu_a, \mu_b) = \frac{\mu_a \mu_b}{\mu_a - \mu_b} \left( e^{-\mu_b \lambda} - e^{-\mu_a \lambda} \right),$$

which is formed by two linearly independent functions (if $\mu_a \neq \mu_b$), $\sim e^{-\mu_b \lambda}$ and $\sim e^{-\mu_a \lambda}$ so $\mu_a$ and $\mu_b$ are identifiable as the unique solution of the identifiability equations

$$\frac{u_{\mu_b} \mu_a u_{\mu_b} \mu_b}{u_{\mu_b} \mu_a - u_{\mu_b} \mu_b} e^{-u_{\mu_b} \mu_b \lambda} = \frac{\mu_a \mu_b}{\mu_a - \mu_b} e^{-\mu_b \lambda}$$

is $u_{\mu_b} = 1$ (because of the exponential). This kind of models need further analysis but they seem to be amenable to our approach.

Finally, while we emphasize the simplicity of the method, it is also amenable to be implemented using symbolic computation packages, particularly for systems with a large number of equations/reactions.

## Supporting information

**S1 Text. In S1 Text we collect the theorems sustaining the method and a catalogue of models with a detailed computation of the identifiability equations that were used to build Table 3.**
(TEX)

## Acknowledgments

This work was initiated during summer visits of the authors to the Los Alamos National Laboratory, and we thank Nick Hengartner and Alan Perelson (LANL) for their hospitality and helpful comments on this work, and the Santa Fe Institute for supporting the summer visits of RdB.

## Author Contributions

**Conceptualization:** Mario Castro, Rob J. de Boer.

**Formal analysis:** Mario Castro, Rob J. de Boer.

**Funding acquisition:** Mario Castro.

**Investigation:** Mario Castro, Rob J. de Boer.

**Methodology:** Mario Castro, Rob J. de Boer.

**Writing – original draft:** Mario Castro, Rob J. de Boer.

**Writing – review & editing:** Mario Castro, Rob J. de Boer.

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
