## [Decision Letter · Decision Letter 0]

2 Jul 2020

Dear Dr Castro,

Thank you very much for submitting your manuscript "Testing structural identifiability by a simple scaling method" for consideration at PLOS Computational Biology. As with all papers reviewed by the journal, your manuscript was reviewed by members of the editorial board and by several independent reviewers. The reviewers appreciated the attention to an important topic. Based on the reviews, we are likely to accept this manuscript for publication, providing that you modify the manuscript according to the review recommendations.

Sincerely,

Miles P. Davenport, MB BS, D.Phil

Associate Editor

PLOS Computational Biology

Douglas Lauffenburger

Deputy Editor

PLOS Computational Biology

[LINK]

Reviewer's Responses to Questions

**Comments to the Authors:**

Reviewer #1: Dear Editor,

The manuscript “Testing structural identifiability by a simple scaling method” focuses on an interesting problem regarding structural identifiability of mathematical models, or the ability to uniquely determine all or some of the parameters or parameter combinations based on observed data.

The paper presents a concise, simple approach to this problem. It is an interesting article that could be of use in the field of mathematical modelling.

I was confused by Table 2, where the authors compared different methods for determining identifiability. Why do different methods arrive at disagreeing results for the same models? Could you explain and clarify how some tests find that the same model is globally structurally identifiable, while others find that the same model is unidentifiable (or locally structurally identifiable)?

I found the examples that were discussed and worked out in the supplementary information very useful, so I think that this information could be included in the main text.

Minor comments:

In the supplement in Section 2.2 in the first sentence, it says “when only x2 is observed”. Should it say “when only x1 is observed”?

Typo right after (18) on p. 5 of supplement. It says, “u_x1)1”but should say “u_x1=1”.

Typo right before section 2.9 on p. 10 of supplement. The “u_d”s should be “u_\\delta”s.

Reviewer #2: the review is uploaded as an attachment

Reviewer #3: See attached

**Have all data underlying the figures and results presented in the manuscript been provided?**

Reviewer #1: Yes

Reviewer #2: Yes

Reviewer #3: Yes

PLOS authors have the option to publish the peer review history of their article (what does this mean?). If published, this will include your full peer review and any attached files.

Reviewer #1: No

Reviewer #2: No

Reviewer #3: No
---

## [Editor Report · Decision Letter 1]

14 Aug 2020

Dear Dr. Castro,

We are pleased to inform you that your manuscript 'Testing structural identifiability by a simple scaling method' has been provisionally accepted for publication in PLOS Computational Biology.

Best regards,

Miles P. Davenport, MB BS, D.Phil

Associate Editor

PLOS Computational Biology

Douglas Lauffenburger

Deputy Editor

PLOS Computational Biology

---

## [Editor Report · Acceptance letter]

20 Oct 2020

PCOMPBIOL-D-20-00674R1 

Testing structural identifiability by a simple scaling method

Dear Dr Castro,

I am pleased to inform you that your manuscript has been formally accepted for publication in PLOS Computational Biology. Your manuscript is now with our production department and you will be notified of the publication date in due course.

With kind regards,

Laura Mallard
